# An Examination of Relative Age and Athlete Dropout in Female Developmental Soccer

**DOI:** 10.3390/sports10050079

**Published:** 2022-05-20

**Authors:** Kristy L. Smith, Patricia L. Weir

**Affiliations:** Department of Kinesiology, University of Windsor, Windsor, ON N9B 3P4, Canada; weir1@uwindsor.ca

**Keywords:** relative age effects, athlete dropout, sport dropout, female, soccer, competition level, sport development

## Abstract

Sport dropout rates among children and youth are a concern for researchers and policy makers. The impact of relative age effects (RAEs) on dropout trends has not been adequately examined in female samples. The purpose of this study was to longitudinally examine dropout in a female soccer cohort in Ontario, Canada. Registration entries for a one-year cohort were examined across a seven-year period (*n* = 9908; age 10–16 years). A chi-square analysis established the presence of RAEs in the initial year of registration. Survival analyses assessed the impact of relative age, competition level, and community size on athlete dropout. A median survival rate of four years was observed for players born in the first quartile, while all remaining quartiles had a median survival of three years. Community size did not predict dropout in this analysis; however, competition level was a significant predictor, with competitive players being more likely to remain engaged vs. recreational players (55.9% vs. 20.7%). The observed trends are likely to have a significant impact from both a healthy development and systems perspective (e.g., economic/market loss). Intervention is needed to mitigate current dropout trends in female athletes. Practical applications are discussed.

## 1. Introduction

Sport dropout rates among children and youth present a growing concern for researchers and policy makers alike. From a healthy development perspective, organized sport participation is associated with a variety of physical, psychological, and social benefits [1,2,3]. For example, youth who engage in organized sport may experience greater social competence [4] and fewer depressive symptoms [5,6], and they may be more likely to develop fundamental movement skills that promote physical engagement in alternative sport contexts and healthy leisure pursuits across their lifespan [7,8]. From a talent development perspective, sport dropout causes a reduction in potential talent for future advancement in sport, as the development of expertise is theoretically predicated by ongoing participation. High rates of organized sport participation have been reported. For instance, seventy-seven percent of Canadian children and youth aged 5–19 years old participate in organized physical activity or sport, as reported by their parents [9]. However, high dropout levels have also been observed and are estimated to be between 30 and 35% per year [10,11], although current estimates are unavailable and likely vary by sex, sport context, and chronological age [12].

A systematic review examining organized sport dropout identified that intrapersonal (e.g., lack of enjoyment) and interpersonal (e.g., parental pressure) constraints are commonly associated with disengagement among children and youth [13]. This review also highlighted a potential connection between frequently cited dropout factors and relative age, that being physical factors (e.g., maturation) and perceptions of competence. Commonly known as Relative Age Effects (RAEs), this term refers to the (dis)advantages resulting from subtle variations in chronological age and thus lived experience and physical/psychological development in age-grouped peers [14]. Within sport, RAEs are believed to advantage those who are relatively older (i.e., born earlier and closer to an organization-imposed cut-off date for grouping similar-age athletes) by providing increased access to higher levels of competition, training, and coaching [15,16].

The underlying mechanisms contributing to RAEs are likely multi-factorial in nature and include a variety of individual, task, and environmental contributors [16,17]; the “maturation-selection” hypothesis is most commonly cited by researchers [18,19,20]. Briefly, this hypothesis suggests that advanced chronological age is accompanied by greater anthropometric (e.g., stature) and physical attributes (e.g., muscular strength and endurance), which provide performance advantages in many sport contexts. These differences are further exacerbated during adolescence. Consequently, relatively older children who are likely to be further along in terms of maturational development receive more attention from coaches and may experience a higher likelihood of selection for elite levels of sport competition, which ultimately furthers their athletic development. Conversely, relatively younger participants may not have the same opportunity to develop and are more likely to struggle with perceptions of competence and self-worth. In Crane and Temple’s review [13], five of the six studies identifying maturation as a contributing factor to dropout suggested that RAEs were a factor, although the reviewers also noted that more research was needed to understand the connection between competing with chronologically older peers and experiences leading to dropout.

Sport popularity has also been associated with RAEs [16,21,22], and this consideration may have a connection to another variable of interest in the sport development literature, that being community size or the “birthplace effect” [23]. Previous research reports have documented increased rates of participation in small to medium-sized communities that are large enough to support youth sport leagues but not so densely populated that the competition for sport facilities, team membership, etc., is detrimental to participation [24,25,26,27]). However, the metrics used to assign community size categories have been somewhat inconsistent, and the role of this variable in sport dropout is unknown. Furthermore, the exact nature of the interaction between community size and RAEs as well as the role these variables play in athlete development outcomes have been somewhat elusive.

Initial observations of an association between relative age and sport dropout were made several decades ago by Barnsley and colleagues [28,29], who suggested that relatively older Canadian ice hockey players were more likely to remain engaged in the sport when compared to the relatively younger participants. Similarly, an examination of male youth soccer in Belgium indicated that a higher dropout rate was present among later-born players at 12 years of age [15]. Large-scale, cross-sectional studies of French soccer and basketball provided further evidence of increased rates of dropout amongst the youngest players over a one-year period [30,31,32]. These trends were consistent across a variety of pre- up to post-adolescent age groups in both male and female samples, leading the researchers to suggest that the over-representation of relatively older participants often observed in sport samples may be in part due to a greater number of relatively younger athletes among the “dropouts” [32].

Two longitudinal examinations of relative age and dropout rate have also been conducted. Figueiredo and colleagues [33] reported the inconsistent tracking of participation by birth quartile for male soccer players at two- and ten-year timepoints after baseline analyses (i.e., at 11 and 13 years of age); playing status could not be predicted by birth quartile. However, this study was limited by a small sample size (*n* = 112). Lemez et al. [34] provided a more substantial analysis of male athletes by examining 14,325 registrants in Canadian ice hockey over a five-year period (age 10–15 years). Relatively younger participants born in the fourth quartile were found to be 17% more likely to drop out than their first-quartile counterparts (OR 1.175, 95% CI 1.054, 1.309). Subsequent analyses attempted to unravel the impact of player movement between competition levels on the observed patterns of dropout. The observations suggested that dropout players were more likely to remain at the same level of competition prior to disengagement from the sport.

While the weight of the evidence in the published literature points to a higher risk of sport dropout for the relatively younger players, one exception to this pattern has been noted. Wattie and colleagues [35] observed increased odds of reported dropout among relatively older female participants at the recreational level in German youth sport clubs, with no comparable effect in the male sample. This finding may have been driven by a high proportion of athletes participating in artistic or individual sport contexts (e.g., gymnastics) within the sample, with smaller physical size providing a competitive advantage. However, these findings also raise questions about the possibility of sex differences in dropout trends. Vincent and Glamser [36] suggested that the “maturation-selection” hypothesis may exemplify the male sporting experience to a greater extent than that of females due to the associated disadvantages that maturation brings to female athletes (e.g., shorter legs and wider hips [37]) as compared to the physical advantages afforded to early maturing males (e.g., increased speed, power, and endurance in motor skills [38]). The findings of Wattie and colleagues may also implicate the role played by talent identification and development processes, as the athletes examined participated in recreational contexts [35]. Indeed, entry into competitive contexts at young ages—known as *early specialization*—has been associated with negative sport experiences (e.g., sport withdrawal, burnout [39,40,41]).

Given the consistent presence of RAEs at the introductory levels and the related evidence with respect to dropout, it is necessary to continue to evaluate participation trends across various age, sport, and competitive levels in a longitudinal manner. Sport participation likely varies across the lifespan, and many factors may contribute to an athlete’s decision to participate in a certain sport context. Consequently, the primary objective of this study was to retrospectively examine dropout in a female cohort across a seven-year period (i.e., covering the pre-adolescent to post-adolescent transition years) with respect to relative age. Given the trends observed in past work that examined sport dropout [30,34] and the consistent reporting of RAEs in soccer (see Smith et al. for a review [42]), it was hypothesized that the relatively older athletes would be more likely to remain engaged in sport across the pre- to post-adolescent years; however, the magnitude could potentially vary based on relevant contextual factors. Thus, additional variables found to influence participation were also evaluated, including community size [24,43] and competition level [34,44].

## 2. Materials and Methods

Following institutional ethics approval, an anonymized dataset of all female members of a one-year cohort registered with *Ontario Soccer* from the age of ten years was obtained from the provincial organization. This dataset included all subsequent registrations across a six-year period for the initial cohort of members (i.e., up to and including existing registration entries at 16 years of age). A total of 38,248 registration entries for 9915 participants were available. Prior to analysis, the participant data were screened for inconsistent and/or missing information with respect to birth month. Twenty-three registration entries were corrected upon confirmation of birth month with a minimum of two other entries for the participant (0.0006% of original sample). One participant was removed because the month of birth could not be confirmed (a total of seven registration entries); one participant was removed because the entries were believed to be a duplicate set (a total of five registration entries); five additional participants were removed because they had an “inactive” status at the age of 10 years and no subsequent registrations beyond that year. Therefore, 99.9% of registration entries were retained (*n* = 9908 participants).

The remaining participants’ birthdates were coded according to birth quartile (i.e., Quartile One-Q1: January—March; Quartile Two-Q2: April—June; Quartile Three-Q3: July—September; Quartile Four-Q4: October—December) in consideration of the December 31st cut-off date employed in Ontario youth soccer. The data were also coded for two other potential determinants of participation. Community size was coded according to *census subdivision.* Census subdivision corresponds to the municipality structure that would determine funding for local sport facilities in Canada [45]. It is a well-established metric used by Statistics Canada and refers to a municipality (as determined by provincial/territorial legislation) or areas treated as municipal equivalents for statistical purposes. Categories that have been employed in previous research were utilized (1: >1,000,000 people; 2: 500,000–999,999; 3: 100,000–499,999; 4: 30,000–99,999; 5: 10,000–29,999; 6: 5000–9999; 7: 2500–4999; 8: 1000–2499; 9: <1000; e.g., [23,25]). 

The level of play at the time of the athlete’s last registration (i.e., competition level prior to disengaging from the sport or at age 16 years) was coded according to the Ontario Soccer organization structure (1: Mini outdoor; 2: Recreational; 3: Competitive). Mini Outdoor is a small-sided game, typically for players 12 years and under. Beyond age 12, players are typically categorized as being at the recreational level (e.g., house league, where selection processes are absent and any child or youth can theoretically participate) or the competitive level (e.g., representative or more elite-level players who gain membership through selection processes or “tryouts”). All registered participants engage in some form of match/game play, although the amount may vary. This structure is recommended by Ontario Soccer and may or may not be followed at the local level (e.g., players may be classified as recreational or competitive prior to age 12 years). These classifications were provided by representatives from Ontario Soccer [46].

A preliminary chi-square analysis and a visual inspection of the birth distribution were conducted to ascertain whether RAEs might be present during the initial year of the registration entries, at the age of ten years. The observed number of participants born in each quartile was compared to the number expected based on the number of days in each quartile. Traditionally, an equal distribution of 25% has been utilized as the expected proportion of participants for each birth quartile in RAE research. Delorme and Champely [47] argue that this method inflates the risk of Type I error. Thus, the actual distribution of the population from which the sample was taken should be utilized, and in the absence of this information, the expected distribution should be adjusted to the number of days present in each birth quartile. For this study, the birth distribution for the overall population of Ontario female soccer players was not available; therefore, the expected distribution was calculated by dividing the number of days in each quartile by 365. A statistically significant chi-square value (*p* < 0.05) was used to calculate the *w* effect size statistic to determine the strength of the relationship. The *w* effect size statistic is calculated by taking the value of chi-square divided by the number of subjects and taking the square root (w = √ (χ2 / n)) [48]. Cohen [48] proposed that *w* values of 0.1, 0.3, and 0.5 represent small, medium, and large effect sizes, respectively. The calculation of standardized residuals was planned for a chi-square analysis producing *w* values ≥ 0.1, with a value of ≥ 1.96 indicating an over-representation and a value of ≤ −1.96 indicating an under-representation in terms of relative age distribution.

Survival analyses were then carried out to assess the impact of relative age on dropout from developmental soccer between the ages of 10 and 16 years. Dropout was identified using the last registration entry that was present in the longitudinal dataset provided by Ontario Soccer. Thus, a participant who last registered at the age of 10 through 15 years would be coded as a “dropout”, and a participant who had a registration entry at the age of 16 years would be coded as “engaged”. A Kaplan–Meier analysis was used to investigate the dropout rate with respect to relative age by birth quartile. The log-rank test assessed the null hypothesis of a common survival curve. This was followed by a Cox Regression to further evaluate the impact of birth quartile, with a consideration of community size and competition level. The proportional hazards assumption was tested using the goodness-of-fit approach [49]. This assumption states that the hazard (i.e., risk of dropping out) for one individual must be proportional to the hazard for any other individual, and that the hazard ratio must be constant over time [49]. 

## 3. Results

### 3.1. General Findings–Relative Age

Results from the preliminary chi-square analysis are presented in Table 1. An over-representation of relatively older participants was observed in the initial sample (χ² (3) = 182.972, *p* < 0.001) with a small effect size (*w* = 0.14). Quartile 2 had the highest number of participants at ten years of age, followed by Q1, Q3, and Q4. The Kaplan–Meier analysis revealed that 23.3% of the initial cohort remained until the end of the seven-year period. The survival curve for each birth quartile is available in Figure 1. The log-rank test indicated that the null hypothesis should be rejected (χ² (3) = 26.321, *p* < 0.001). A median survival rate of four years was observed for players born in the first quartile over the subsequent six years of registration; this differed from a median survival of three years for players born in the remaining quartiles (outlined further in Table 2).

### 3.2. Additional Factors—Competition Level and Community Size

Prior to conducting the Cox Regression, it was recognized that players who dropped out during “mini outdoor” would bias the survival analysis as any player who was classified in this category (i.e., coded according to last registration entry) would theoretically drop out by the age of 12 years, according to Ontario Soccer’s organizational structure. Thus, only players coded as “competitive” (*n* = 2327) and “recreational” (*n* = 4836) at the time of their last registration were included in the Cox Regression (overall *n* = 7163). The findings are presented in Table 3. The analysis indicated that birth quartile was not statistically significant (*p* > 0.05) when the impact of community size and competition level were considered. Community size did not predict dropout in this analysis; however, competition level was observed to be a significant predictor of continued sport involvement (*p* < 0.001).

The survival and hazard functions using the mean for competition level can be found in Figure 2a,b, respectively. By percentage, 55.9% of competitive players were still registered with Ontario Soccer at the age of 16 years, while only 20.7% of recreational-level players remained (see Table 4 and Figure 2a). Descriptively, this corresponds to a yearly dropout rate of more than 30% of recreational players each year. Competitive players were more than twice as likely to remain engaged in soccer until the age of 16 years when compared to recreational-level participants (Hazard ratio 2.593, 95% CI 2.419, 2.779; see Figure 2b). In consideration of the significance of competition level, a graphical representation of the quartile distributions for each year was generated for both the competitive and the recreational streams to inspect the transient relative age distribution. The competitive trajectory (see Figure 3a) showed a classic RAE, with Quartile 1 consistently over-represented and Quartile 4 consistently under-represented across the seven-year period; on the other hand, the recreational stream (see Figure 3b) showed an over-representation in Quartile 2 and an under-representation in Quartile 4.

## 4. Discussion

The primary objective of this study was to retrospectively examine athlete dropout with respect to birth quartile in a female cohort for a total of seven years: beginning at the age of ten years, and subsequently followed across a six-year period. Thus, this study provides a longitudinal snapshot of the pre-adolescent to post-adolescent transition years within female soccer in Ontario. A significant RAE was observed in the initial cohort with the relatively oldest participants (i.e., those born earlier in the same-age cohort) having the highest rates of participation at age ten years. The participants born in the first quartile were found to have a greater likelihood of continued engagement in youth soccer during the examination period, as inferred by a median survival rate of one additional year when compared to their peers. However, birth quartile was not found to be a significant factor when competition level and community size were considered as part of the analysis. Thus, the preliminary hypotheses were generally supported by the results of the analyses.

The outcome of this study suggests that female dropout patterns in Ontario Soccer are comparable to previous findings in team sport contexts, with the relatively youngest exhibiting higher rates of disengagement. The one noted exception in the literature [35] may be differentiated by the artistic/individual sport contexts in which the participants engaged. Physical contact is inherent in the sport of soccer, providing an advantage to those with advanced growth and/or maturational status. Additionally, the team context might also emphasize physical differences as comparisons between players occur on the field and are generally based on more subjective evaluations of participants by coaches as opposed to objective measures that are more commonly associated with individual sports (e.g., a 100-meter swim time [50]). The aforementioned sample [35] was also considered to be “recreational” in nature. Interestingly, competitive level was observed to be an important variable in the current analysis, negating the impact of birth quartile when included in the analysis.

If considered to be an accurate estimate, the findings of this study suggest that approximately 7200 participants (or 73%) of this one-year, provincial cohort (*n* = 9908) are at risk of dropping out one year earlier because of their birthdate position with respect to an arbitrary, age-group cut-off. This statistic is alarming from both a healthy development perspective (i.e., continued participation is associated with positive outcomes; see examples discussed in the Introduction) and a systems perspective (i.e., continued growth of the sport). For example, a significant reduction in participation contributes to an economic/market loss [51]; that is, a high rate of dropout contributes to a reduction in game interest, loss of membership fees, and a reduced talent pool for future advancement in sport. Furthermore, youth sport is predominantly run by volunteers. Individuals who disengage from a sport during childhood or adolescence may be less likely to transition to a contributive role in their adult years.

These findings also highlight the potential impact of competitive streaming on sport dropout. While a greater proportion of competitive-level players were engaged at the age of 16 years (55.9% vs. 20.7% for recreational-level players), a more biased birthdate distribution favoring the relatively older players was also evident in the competitive context when evaluated by year of registration (see Figure 3a). This may suggest that RAEs resulting from initial growth differences are being perpetuated by talent selection processes [42], and is consistent with the available research examining female youth soccer players [52,53]. At no point during the seven-year period were the relatively youngest athletes observed to “catch-up” despite the culmination of maturational processes within the examined timeframe. While the recreational stream had a more evenly distributed birth representation (see Figure 3b), the high disengagement of athletes over the seven-year period may highlight a concerning trend for recreation-level athletes. This is somewhat surprising given the reduced demands of playing at the recreational level as compared to higher levels of competition, where the increased demands of additional training and performance might conflict with other priorities for this age demographic (e.g., schoolwork, part-time employment, social activities). However, it may also be indicative of athletes choosing to prioritize alternative forms of sport participation.

Community size did not appear to be a significant factor with respect to sport dropout in this sample. This finding differs from previous research studies (e.g., [24,25,26,27]) that have found increased rates of participation in small to medium-sized communities that are large enough to support youth sport leagues but not so densely populated that the competition for sport facilities, team membership, etc., is detrimental to participation. The survival analyses utilized in this study may not have detected subtle trends related to sport dropout in this sample due to the large range of community sizes in Ontario (i.e., census subdivisions range from 5 to 2,615,060 inhabitants). The impact of community size in this sample is evaluated further in a separate study that used geospatial mapping and odds ratio analyses [54].

Although not a primary goal of this work, this study documented the over-representation of the second quartile in the initial cohort at ten years of age (followed by Q1, Q3, and Q4); this provided the first RAE observed in a Canadian soccer sample. This pattern differs from the classic, linear RAE pattern (Q1 > Q2 > Q3 > Q4) that would be expected, based purely on chronological age differences. Female samples have been associated with a Q2 over-representation in previous studies, particularly in Canadian ice hockey at developmental and national levels [44,55]; but also observed in post-adolescent [56] and adult [22,30] female soccer samples.

The cause of this Q2-trend has largely been undetermined to date. Previous hypotheses have suggested that the “best” Q1-born, female athletes may be playing in male sport to gain a competitive advantage or are perhaps engaged in a more popular sport, leaving those born in the second quartile to experience success in the context under examination. This study adds evidence against the latter hypothesis in consideration of the Canadian Heritage Sport Participation 2010 report [57], which identified soccer as the most highly played sport by Canadian children. However, it was noted that the Q2 over-representation in this study was primarily driven by registration numbers in the recreational context when the sample was evaluated according to the competitive stream (cf. Figure 3a vs. Figure 3b), suggesting that the relatively oldest were experiencing greater success within the context of soccer at both the competitive and recreational levels.

Underlying patterns observed in a sample compiled for a recent meta-analysis of female athletes provide evidence that the effect might possibly be associated with early specialization opportunities for Q1-born athletes and consequent burnout, injury, and/or sport withdrawal (see Smith et al. [42] for further discussion). This hypothesis might partially explain the observed trends in this sample. However, the birth quartile distribution showed essentially the same pattern of representation across all years examined at both the competitive (i.e., Q1 over-representation) and the recreational (i.e., Q2 over-representation) levels; no transitional RAEs were observed. Thus, the underlying mechanisms of these trends requires further examination, and the exact contributor in this sample and others remains unknown.

The dropout rates observed in this longitudinal analysis are reflective of the high rates of dropout that have been observed in other samples (e.g., [10,11,58,59]). Sport administrators should seek to organize sport in a way that promotes the personal development of all its members, with varying levels of ability and motivation [3,60]. Strategies that support recreational-level athletes appear to be particularly needed. Future applied research should evaluate whether the provision of opportunities for skill development and other experiences that competitive players have (e.g., tournaments, inter-city play, skill development initiatives, team building events) would encourage engagement in recreational streams with increasing chronological age while still maintaining the reduced time demands (vs. competitive levels) that are likely to be desirable for high-school-age athletes. The recent trend towards sport-specific academies (i.e., academic institutions offering combined athletic and academic curricula) may be a promising avenue for continued sport engagement into the adolescent/post-adolescent years as they offer access to facilities/coaching and a flexible academic schedule. However, continued alignment between these academies and existing sport governing bodies is needed [61,62].

This study adds to the limited pool of research on female soccer athletes. A review of female RAEs found small but consistent RAEs in this sport context [42]; however, the existing work has primarily focused on elite competitors in post-adolescent and adult age groups (see Smith et al. [42] for a review) as opposed to the more developmental levels of the sport. Thus, the documentation of RAEs in pre-adolescent/adolescent and recreational-level athletes is important. The majority of work in soccer has focused on RAEs for male athletes [22,63], and additional examinations of RAEs for females at all levels of the sport are needed in order to inform meaningful interventions that reduce the inequities in athlete development. Furthermore, this study adds to the limited literature available that examines relative age and dropout in a longitudinal manner within a youth sport sample. To date, dropout from organized sport with respect to relative age has not been adequately studied, and a continued evaluation of the patterns that exist in different sport contexts (i.e., team vs. individual, competitive vs. recreational), across age groups, and between the sexes is required. Following a one-year cohort through the pre-adolescent to post-adolescent transition was an important element in this analysis, as adolescence has been identified as a critical timepoint for overall declines in physical activity levels [64]. However, information is still lacking with respect to participants who declined participation prior to the age of ten years and beyond 16 years of age. An evaluation of a broader age range and a comparative male sample from Ontario youth soccer would be beneficial.

Future studies also need to consider the longitudinal nature of sport participation along with the dynamic nature of athletic development. For instance, Cobley et al. [65] identified transient relative age advantages among national-level Australian swimmers, with the relatively oldest and youngest being over-represented at different time points (i.e., age 12 and 18 years, respectively); this suggests that detailed examinations to increase knowledge and understanding of relative age mechanisms are justified. A multi-level systems perspective should be maintained [66,67,68] in these future investigations, as athlete development does not occur within a *vacuum* [17].

The use of survival analysis provided an alternative way for assessing dropout, that being the use of time-to-event data. Traditional statistical methods of assessing the birth date distributions of athlete samples, such as chi-square analysis and linear regression, cannot handle the censoring of events (i.e., when survival time is unknown). However, as discussed above, a survival analysis may not be sensitive enough to pick up community size-related variations, and this variable will require a deeper level of examination in future studies. A consistent approach was taken to the coding of each participant’s registration entry by census subdivision due to the correlation of this variable with municipal funding for sport facilities; this consistency was lacking in previous research on community size. However, this approach still has limitations as the census subdivision may not be the true size of the community and does not account for the proximity of neighboring communities, which might provide additional options for sport club membership, opportunities for training, an enhanced pool of competition, etc. Finally, this analysis is limited in the same manner as many studies using relative age; the evaluation of quantitative trends cannot answer the questions of “why” and “how” relative age influences dropout. Mixed-method approaches and person-centered analyses are needed in future research to learn more about the athletes on an individual level [17]. Researchers should seek to gain a better understanding of the developmental experiences of individuals who succeed despite a disadvantageous relative-age position within a cohort in order to inform sport engagement strategies and promote positive sport experiences and the equitable distribution of opportunities for all athletes.

## 5. Conclusions

Relative age effects are present in developmental-level, female soccer in Ontario. A higher risk of dropout is incurred by the relatively youngest and recreational-level players. Future research is needed to confirm the exact mechanism(s) contributing to these trends and to determine effective methods of supporting at-risk athletes.

## Figures and Tables

**Figure 1 sports-10-00079-f001:**
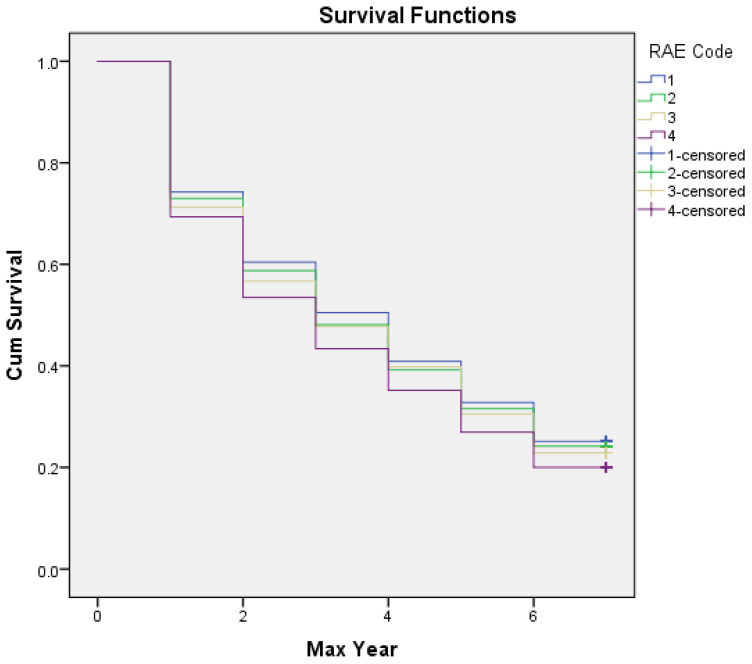
Survival curve for each birth quartile, indicating the highest cumulative survival over the seven-year period.

**Figure 2 sports-10-00079-f002:**
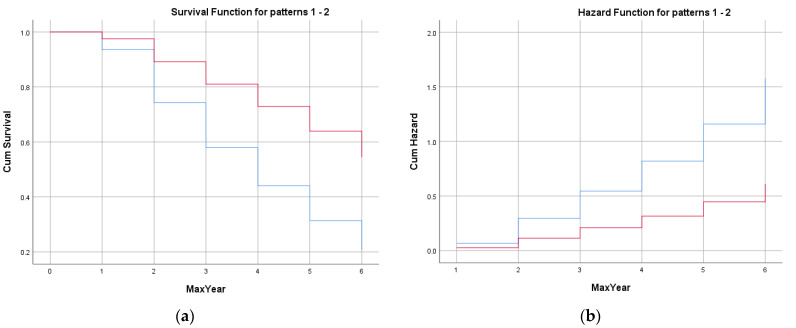
Competitive (red) and recreational (blue): (**a**) Survival function at the mean for the competition level. The vertical axis shows the probability of survival. The horizontal axis represents time-to-event data. (**b**) Hazard function at the mean for the competition level. The vertical axis shows the cumulative hazard, equal to the negative log of the survival probability. The horizontal axis represents time-to-event data.

**Figure 3 sports-10-00079-f003:**
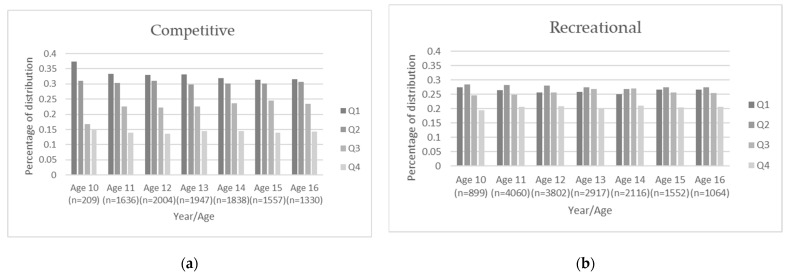
Birth distribution by quartile and chronological age (10–16 years). (**a**) Competitive players; (**b**) Recreational players. Note: The majority of participants ages 10–12 years would be classified as “mini outdoor” according to Ontario Soccer’s organizational structure and are therefore not represented.

**Table 1 sports-10-00079-t001:** Results from the preliminary chi-square analysis.

Birth Quartile	Observed (*n*)	Expected	Standardized Residual	Odds Ratio	95% CI	
					Lower	Higher
Quartile 1 (Q1)	2674	2443.1	**4.56**	**1.13**	1.07	1.19
Quartile 2 (Q2)	2803	2470.2	**6.74**	**1.19**	1.13	1.25
Quartile 3 (Q3)	2472	2497.4	−0.476	0.99	0.92	1.05
Quartile 4 (Q4)	1959	2497.4	**−10.745**	**0.73**	0.67	0.80

Note: Values in bold indicate an over-representation (i.e., ≥ 1.96) or under-representation (i.e., ≤ −1.96) with respect to relative age distribution by quartile.

**Table 2 sports-10-00079-t002:** Results from the Kaplan–Meier survival analysis: mean and median values for survival time.

	Mean				Median			
Birth Quartile	Est.	Std. Error	95% CI		Est.	Std. Error	95% CI	
			Lower	Upper			Lower	Upper
Q1	3.840	0.046	3.750	3.929	4.000	0.097	3.809	4.191
Q2	3.748	0.045	3.660	3.835	3.000	0.097	2.811	3.189
Q3	3.688	0.048	3.595	3.782	3.000	0.119	2.767	3.233
Q4	3.483	0.052	3.381	3.586	3.000	0.086	2.831	3.169
Overall	3.705	0.024	3.659	3.752	3.000	0.054	2.894	3.106

Note: Estimation is limited to the largest survival time if it is censored.

**Table 3 sports-10-00079-t003:** Results from the Cox Regression survival analysis (overall).

	Regression Coefficient	Std. Error	*p* > |z|	Hazard Ratio	95% CI	
					Lower	Upper
Q1	0.015	0.043	0.717	1.016	0.934	1.104
Q2	0.005	0.042	0.901	1.005	0.926	1.092
Q3	0.025	0.043	0.565	1.025	0.942	1.116
CS	0.003	0.002	0.080	1.003	1.000	1.007
Comp. Level	0.953	0.035	0.000	2.593	2.419	2.779

Notes: Quartile 4 used as reference category. Community size (CS) divided by 100,000 for analysis purposes. Confidence intervals that include a value of 1.0 indicate equivalence in the hazard rate (i.e., not statistically significant).

**Table 4 sports-10-00079-t004:** Results from the Cox Regression survival analysis (competition level).

Competitive Level	Dropout before Age 16 Years (*n*)	Engaged at Age 16 Years (*n*)	Engaged at Age 16 Years (%)	Overall *n*
Competitive	1027	1300	55.9%	2327
Recreational	3835	1001	20.7%	4836

## Data Availability

The data analyzed in this study were obtained from Ontario Soccer. Access to the data is not possible due to ethical considerations as they contain personal information.

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
