# Peer review of "An Examination of Relative Age and Athlete Dropout in Female Developmental Soccer"

_sports, 2022, doi:10.3390/sports10050079_

Round 1
Reviewer 1 Report
This manuscript reports dropout rates among female soccer players aged 10 to 16 in Ontario, Canada. The authors confirm the relative age effect at entry into the registry of Ontario Soccer. Continued engagement in soccer is examined by survival analysis. The authors show that the relative age effect persists throughout the observation period, especially in the “competitive” level.
Major comments
- The manuscript is nicely written, but the introduction is exceedingly long. It is more of a review of the relative age effect than an introduction to the subjects studied. It can be abbreviated considerably and parts of the introduction fit better in the discussion.
- The authors separate between a “competitive” and a “recreational” level. These levels need to be better defined. Are no matches played at the “recreational” level? Are all “recreational” players necessarily registered with Ontario Soccer? In other words, can we be sure that the dropouts have abandoned sport?
- A major advantage of the study is the large sample size. A major disadvantage is that very little information is available about the subjects. There are hypothetically a large number of social factors that can affect dropout rate from sport. The authors analyse community size, which seems like a rather blunt instrument. What about socio-economic differences between communities? Parents’ engagement in sport? Parents’ level of education and employment? Distance to training facilities?
Author Response
Thank you for your thoughtful review of our manuscript! We have outlined the changes in the attached response document.

Reviewer 2 Report
General Comments
This manuscript examines the relative age effect (RAE) within (n = 9,908 female soccer players, paying particular attention to drop-outs. This this study is of interest to better understand the RAE phenomenon in elite female soccer and will contribute to current knowlege. However, in the current version, I have some minor concerns. In the following, I have listed my suggestions.
Minor suggestions/concerns:
Intro:
- there are some current studies analysing RAEs in female soccer. Please integrate them in your introduction/and or discussion.
Results:
- please provide odds ratios and confidence intervals for the description of RAEs
- please discuss "the underdog hyposesis" which is present in male soccer, but seems to be absent in female soccer
Author Response

(The authors gave the same response as above.)
